# Castration modulates singing patterns and electrophysiological properties of RA projection neurons in adult male zebra finches

Songhua Wang, Congshu Liao, Fengling Li, Shaoyi Liu, Wei Meng and Dongfeng Li

School of Life Science, South China Normal University, Key Laboratory of Ecology and Environmental Science in Higher Education of Guangdong Province, Guangzhou, China

## ABSTRACT

Castration can change levels of plasma testosterone. Androgens such as testosterone play an important role in stabilizing birdsong. The robust nucleus of the arcopallium (RA) is an important premotor nucleus critical for singing. In this study, we investigated the effect of castration on singing patterns and electrophysiological properties of projection neurons (PNs) in the RA of adult male zebra finches. Adult male zebra finches were castrated and the changes in bird song assessed. We also recorded the electrophysiological changes from RA PNs using patch clamp recording. We found that the plasma levels of testosterone were significantly decreased, song syllable's entropy was increased and the similarity of motif was decreased after castration. Spontaneous and evoked firing rates, membrane time constants, and membrane capacitance of RA PNs in the castration group were lower than those of the control and the sham groups. Afterhyperpolarization AHP time to peak of spontaneous action potential (AP) was prolonged after castration. These findings suggest that castration decreases song stereotypy and excitability of RA PNs in male zebra finches.

## INTRODUCTION

Steroid sex hormones change oscines song behavior and modulate the underlying neural substrates in adulthood. Androgens, particularly testosterone, play important roles in stabilizing birdsong. For example, testosterone can increase the stability of song (*Meitzen et al., 2009a*; *Sizemore & Perkel, 2011*), the size of song nuclei (*Brenowitz et al., 1998*; *Brenowitz et al., 1991*; *Dloniak & Deviche, 2001*; *Hall & Macdougall-Shackleton, 2012*; *Meitzen & Thompson, 2008*), the expression of androgen receptor mRNA (*Fraley et al., 2010*; *Wacker et al., 2010*) and the excitability in song-control neurons (*Meitzen et al., 2007a*; *Meitzen, Perkel & Brenowitz, 2007b*) in seasonally breeding songbirds.

The robust nucleus of the arcopallium (RA) is a crucial nucleus in the song control system, receiving synaptic input from both the HVC (used as a proper name) and lateral

Corresponding author
Dongfeng Li, dfliswx@126.com

magnocellular nucleus of the anterior nidopallium (LMAN) (*Kao, Wright & Doupe, 2008*; *Olveczky, Andalman & Fee, 2005*; *Wild, 1993*). The HVC-RA pathway produces a stereotyped pattern contributing to stable song, and while the LMAN-RA pathway fires when male birds sing to female birds, LMAN neurons exhibit reliable firing of single spikes precisely locked to song. Thus, the LMAN may act as a source of variability (*Nottebohm, 2005*). The axons of projection neurons (PNs) in the ventral RA project topographically onto the hypoglossal motor nucleus (nXIIts) that innervates the syrinx, and the axons of PNs in the dorsal RA project to the areas in the lateral medulla that control respiration (*Vicario, 1994*; *Xie et al., 2010*). Lesion of the RA causes severe song deficits (*Nottebohm, Stokes & Leonard, 1976*). Moreover, RA activity is significantly correlated with variations in the spectral entropy of syllables (*Sober, Wohlgemuth & Brainard, 2008*), and RA shows accurately timed and structured bursts of activity that are associated with specific syllables (*Yu & Margoliash, 1996*).

Testosterone and estradiol act on brain areas to affect a multitude of attributes, including the rate of neuronal survival and excitability in song-control neurons (*Meitzen et al., 2007a*; *Meitzen & Thompson, 2008*; *Thompson, 2011*; *Thompson & Brenowitz, 2005*; *Thompson & Brenowitz, 2010*). The HVC and area X express androgen/estrogen receptors, however, other nuclei of the song control system only express androgen receptors (*Meitzen et al., 2009b*). Androgen and estrogen receptors in the HVC can be co-activated to drive the change of the excitability of RA PNs, thus testosterone and estradiol can act transsynaptically within the HVC and RA to regulate neuronal phenotype (*Meitzen et al., 2007a*). Recent studies have shown that testosterone and photoperiod affect the excitability of RA PNs in seasonally breeding songbirds that undergo major hormonal shifts as a result of photoperiod, but have no effect on the electrophysiological properties of HVC neurons (*Meitzen et al., 2009b*). Zebra finches are opportunistic breeders rather than seasonal breeders (*Prior, Heimovics & Soma, 2013*). Castration of adult male zebra finches declines testosterone levels in plasma (*Adkins-Regan et al., 1990*), reduces their singing rate but has no effect on the structure of song (*Arnold, 1975*). Castration and antisteroid treatment extremely disrupts the stereotypy and acoustic quality of individual note production, as well as stereotypy of the temporal structure of the song phrase in juvenile male zebra finches, but exerts no deleterious effects on the quality of song behavior in adulthood (*Bottjer & Hewer, 1992*). In adult male zebra finches, high levels of testosterone decrease the frequency of syllable in song and reduce the potential for vocal plasticity (*Cynx, Bean & Rossman, 2005*; *Williams, Connor & Hill, 2003*), we predicted that the decline of testosterones levels may induce the plastic song. RA is a crucial nucleus for singing pattern. It is unknown whether the change in singing pattern is accompanied by changes in electrophysiological properties of RA in adult male zebra finches. To address these issues, we examined the effect of castration on singing patterns and electrophysiological properties of RA PNs in adult male zebra finches. Our data showed that castration decreases song stereotypy and excitability of RA PNs.

## MATERIALS AND METHODS

### Animals and experimental treatments

A total of 27 adult male zebra finches (*Taeniopygia guttata*) (>120 days old) obtained from a commercial breeder were used in this study. All experiments were carried out in accordance with the University and China animal guidelines. The care and use of animals for this study was approved by the Institutional Animal Care and Use Committee at South China Normal University and in accordance with National Institutes of Health guidelines (scnu20070033). Birds were housed in stainless steel cages ($23.5 \times 22.5 \times 27.5$ cm), and each of the cages contained a pair of male and female birds, which were provided with ad libitum food and water and were kept in 14:10 h light/dark cycles. All birds were divided into three main experimental groups: castration group ($n = 11$ birds), control group ($n = 12$ birds), and sham group ($n = 4$ birds).

Before castration, the songs of all birds were recorded in the presence of adult female birds. Birds were then anesthetized with 10% chloral hydrate (0.02 mL/10 g). A small incision was made on the lateral wall of the body cavity between the last two ribs just ventral to the ventral margin of the kidney (*Arnold, 1975*). The testicles were removed with ophthalmic forceps. The sham group underwent the same surgery without removing the testicles. The control group did not receive surgery.

### Song recording

The song recording room ($2.5 \times 2 \times 2.5$ m) contained TAKSTAR directional microphones (Guangdong Victory Electronics Co. Ltd., Guangzhou, China; frequency range: 50–20000 Hz) and a glass window ($85 \times 65$ cm). Birds in the song recording room could see the other birds from the glass window. When the songs were recorded, the male bird was placed in a cage in the song recording room near the glass window, while the female bird was placed in a cage near the glass window outside of the song recording room, so that the male bird could observe the female bird through the window. On each recording day, every bird was recorded for 90–120 min. Songs were recorded between 8:00 a.m. and 11:00 a.m. Songs were recorded before the castration and sham operation. When birds produce a stable song, the date defined as 'pre'. The songs were then recorded again at the 30th day after castration and sham operation. The songs of birds in control group also were recorded at 'pre' and 30th day. Song recording was performed using Cool Edit 2000 (Adobe Systems Co., SAN Jose, CA, USA; sampling rate: 44100 Hz; channels: stereo; resolution: 16-bit).

### Stereotypy of song

We analyzed song stereotypy by calculating entropy (a measure of randomness, entropy is high when the waveform is random, and low when the waveform is of pure tone) (*Meitzen et al., 2009a*; *Tchernichovski et al., 2000*) of the longest syllable (the distance-call element, whose structure matched that of distance call and is derived from distance call *Zann, 1996*) in the motifs within a record using Sound Analysis Pro 2011 (contrast: 0, frequency range: 0–11025 Hz, FFT data window: 10 ms, advance window: 1 ms, contour thresh: 10).

On each recording day the entropy of 30 syllables in 30 motifs was analyzed. Sixty motifs were used to analyze the percentage similarity (% similarity) of the motif in the song (*Meitzen et al., 2009a*; *Sizemore & Perkel, 2011*). Higher entropy indicates less stereotypy, while higher (% similarity) indicates more stereotypy.

## Slice preparation

At the 30th day after castration, the birds were anesthetized with 10% chloral hydrate and then rapidly decapitated. Brains were dissected into ice-cold, oxygenated (95% $O_2$ and 5% $CO_2$) slice solution. Slice solution consisted of KCl 5 mM, $NaH_2PO_4 \cdot H_2O$ 1.26 mM, $MgSO_4 \cdot 7H_2O$ 1.3 mM, $NaHCO_3$ 28 mM, glucose 10 mM, sucrose 248 mM, and NaCl 62.5 mM (*Bottjer, 2005*). Coronal brain slices (250–300 μm thick) containing the RA were cut with a vibrating microtome (World Precision Instruments Inc., Sarasota, FL, USA) and collected in artificial cerebrospinal fluid (ACSF) that was warmed to 37 °C. After 30 min the ACSF was cooled to 35 °C, and the slices were allowed to recover in the holding chamber for 1–1.5 h. Standard ACSF consisted of NaCl 125 mM, KCl 2.5 mM, $NaH_2PO_4 \cdot H_2O$ 1.27 mM, $MgSO_4 \cdot 7H_2O$ 1.2 mM, $NaHCO_3$ 25 mM, glucose 25 mM and $CaCl_2$ 2.0 mM, and the osmolality was adjusted with sucrose to 350 mOsm (*Bottjer, 2005*).

## Patch-clamp recording

During the experiments, slices were transferred to a recording chamber where they were continuously perfused with ACSF, saturated with 95% $O_2$ and 5% $CO_2$ at room temperature (23–28 °C). RA and the surrounding tissues were observed at low magnification (50×) under a BX51WI microscope connected with a DIC-IR video camera (Olympus, Tokyo, Japan). At high magnification (400×), RA neurons were visualized and the recordings were made from RA PNs. Recording pipettes were fabricated from borosilicate glass (Sutter Instrument Co., Novato, CA, USA) using a Flaming-Brown puller (Micropipette Puller P-97; Sutter Instrument Co.), and were filled with the solution containing $KMeSO_4$ 120 mM, NaCl 5 mM, HEPES 10 mM, EGTA 2 mM, Mg-ATP 2 mM, and Na-GTP 0.3 mM (pH 7.3–7.4). Osmolality was adjusted with sucrose to 340 mOsm. The recording pipettes, which had resistances ranging from 4 to 7 MΩ, were positioned using an integrated motorized control system (Sutter Instrument Co.). Signals were amplified with a MultiClamp 700B (Axon Instruments, Sunnyvale, CA, USA). Signals were low-pass filtered at 5 kHz, digitized at 10 kHz with DIGIDATA 1322A (Axon Instruments) and acquired using Clampfit 9.2 (Axon Instruments). Tight-seal and whole-cell recordings were obtained using standard techniques. The baseline membrane potential was held at −70 mV during the stimulation protocols. RA PNs were identified by their distinct intrinsic properties as described previously (*Spiro, Dalva & Mooney, 1999*).

## Electrophysiological data analysis

Clampfit 9.2 and Origin Pro 8.0 (Origin Lab, Northampton, MA, USA) were used for analysis. In measuring spontaneous firing rates in the cell-attached configuration, we analyzed the spike amplitude, waveform, and time derivative to ensure that spike events

were single units. We measured spontaneous activity for at least 5 min, and calculated the firing rate by dividing the number of spikes observed by the duration of the recording as reported (*Meitzen et al., 2007a*; *Meitzen, Perkel & Brenowitz, 2007b*). Action potentials (AP) were detected using the event detection package of the Clampfit 9.2. Spontaneous firing rates were calculated at the beginning of the recording as soon as it stabilized following patch rupture. The AP threshold was detected using a custom algorithm described previously by Baufreton (*Baufreton et al., 2005*); the afterhyperpolarization (AHP) peak amplitude was the difference between the AP threshold and the most negative voltage reached during the AHP. The AHP time to peak was the time of this minimum minus the time when the membrane potential crossed the AP threshold on descent from the AP peak (*Farries, Meitzen & Perkel, 2005*). For each neuron, the measurements of five APs were averaged to produce the final AP measurements for that neuron. Evoked firing rates were measured after patch rupture. The evoked firing rate was defined as the number of AP evoked over the duration of the current injection. The slope of the F–I relationship was estimated by linear fitting. Slope parameters were estimated separately for individual neurons and mean slope values were averaged for the whole groups of neurons. Input resistance was estimated by applying small hyperpolarizing current pulses. The membrane time constant was calculated by fitting a single exponential curve to the membrane potential change in response to $-200$ pA hyperpolarizing pulses. Membrane capacitance was calculated using the following equation: capacitance $=$ membrane time constant/input resistance (*Meitzen et al., 2009b*).

## Hormone assay

On the day of each electrophysiological recording, carotid artery blood was rapidly collected from each subject before they were decapitated into a heparinized microhematocrit tube and stored on ice until centrifugation (within 1 h). We did not double check that the testes were totally removed after animal sacrifice. The plasma was harvested and stored at $-80\,°C$. To measure circulating testosterone levels, enzyme-linked immunosorbant was used in a bird testosterone ELISA kit (IBL, Hamburg, Germany), which contained a substrate standard. The minimum detectable plasma testosterone concentration was 0.12 ng/mL, and the maximum was 7.20 ng/mL. All samples were tested in one single assay.

## Statistical analysis

All values are reported as mean $\pm$ SEM. We used two-way repeated measures ANOVA to compare the song data at the 30th day after castration and sham operation with the song data at 'pre' (see the part of song recording), and injected current on the evoked firing rate of RA PNs in the castration group with sham and control groups. We used one-way ANOVA to compare the song data at the 30th day with the song data at 'pre' in the control group. We also used one-way ANOVA to compare all plasma testosterone levels and other electrophysiological data of RA PNs in the castration group with sham and control groups. *P* values $< 0.05$ were considered significant.

## RESULTS

### Plasma testosterone levels

In the castration group ($n = 11$), plasma testosterone levels were lower ($3.91 \pm 0.08$ ng/mL) compared with the control group ($n = 12$) ($5.15 \pm 0.08$ ng/mL, $F_{(1,22)} = 150.49, P < 0.01$) and the sham group ($n = 4$) ($5.27 \pm 0.09$ ng/mL, $F_{(1,13)} = 98.32, P < 0.01$).

### Stereotypy of the song before and after castration

We randomly selected five birds in each of the castration and control groups, respectively, to analyze the stereotypes of their songs, while four birds were analyzed in the sham group. Zebra finch song usually contains motifs. Every motif includes two to eight syllables (*Nordeen & Nordeen, 2010*). In our experiment, we recorded song motifs from castration (Figs. 1A$_1$ and 1A$_2$), control and sham (Figs. 1B$_1$ and 1B$_2$) groups, and analyzed the entropy of syllable and % similarity of the motif in each group (Fig. 2).

The longest syllable of motif was first analyzed. In the castration group, the entropy was altered gradually, and entropy was significantly increased from $-3.71 \pm 0.31$ to $-3.31 \pm 0.33$ ($F_{(1,58)} = 33.61, P < 0.01$) at the 30th day after castration (Fig. 2A). To test the effect of castration on all syllables in the motif, we analyzed other syllables, as shown in Figs. 1A$_1$ and 1A$_2$. The entropy of syllable 'a' in 'pre' was $-2.76 \pm 0.06$, while at the 30th day after castration the entropy changed to $-2.48 \pm 0.05$. Castration increased the entropy of syllable 'a' ($F_{(1,58)} = 10.77, P < 0.01$). Castration also increased the entropy of other syllables (Figs. 1A and 1B).

Next, we analyzed the % similarity of the motif. Castration significantly decreased the % similarity of the motif from $93.83 \pm 0.80$ to $87.7 \pm 1.04$ ($F_{(1,58)} = 201.32, P < 0.01$) (Fig. 2B). However, in the sham group the entropy of syllables and % similarity of the motif did not change before and after operation, and were similar to the control group (Figs. 2A and 2B).

### Electrophysiological properties of RA PNs

21 RA PNs from 11 birds of the castrated group, 23 RA PNs from 12 birds of the control group, and 8 RA PNs from 4 birds of the sham group were recorded.

### Castration decreased spontaneous firing rates in the cell-attached configuration

When cells were sealed, many RA PNs were spontaneously active *in vitro*, as described previously in wild song sparrows (*Meitzen, Perkel & Brenowitz, 2007b*). Castration significantly affected the spontaneous firing rate of RA PNs compared with the control group ($F_{(1,42)} = 7.85, P < 0.01$) and the sham group ($F_{(1,19)} = 8.41, P < 0.01$) (Table 1, Figs. 3A–3C). The mean firing rate of RA PNs was approximately 1.5 times higher in the control group and sham group than that in the castrated group.

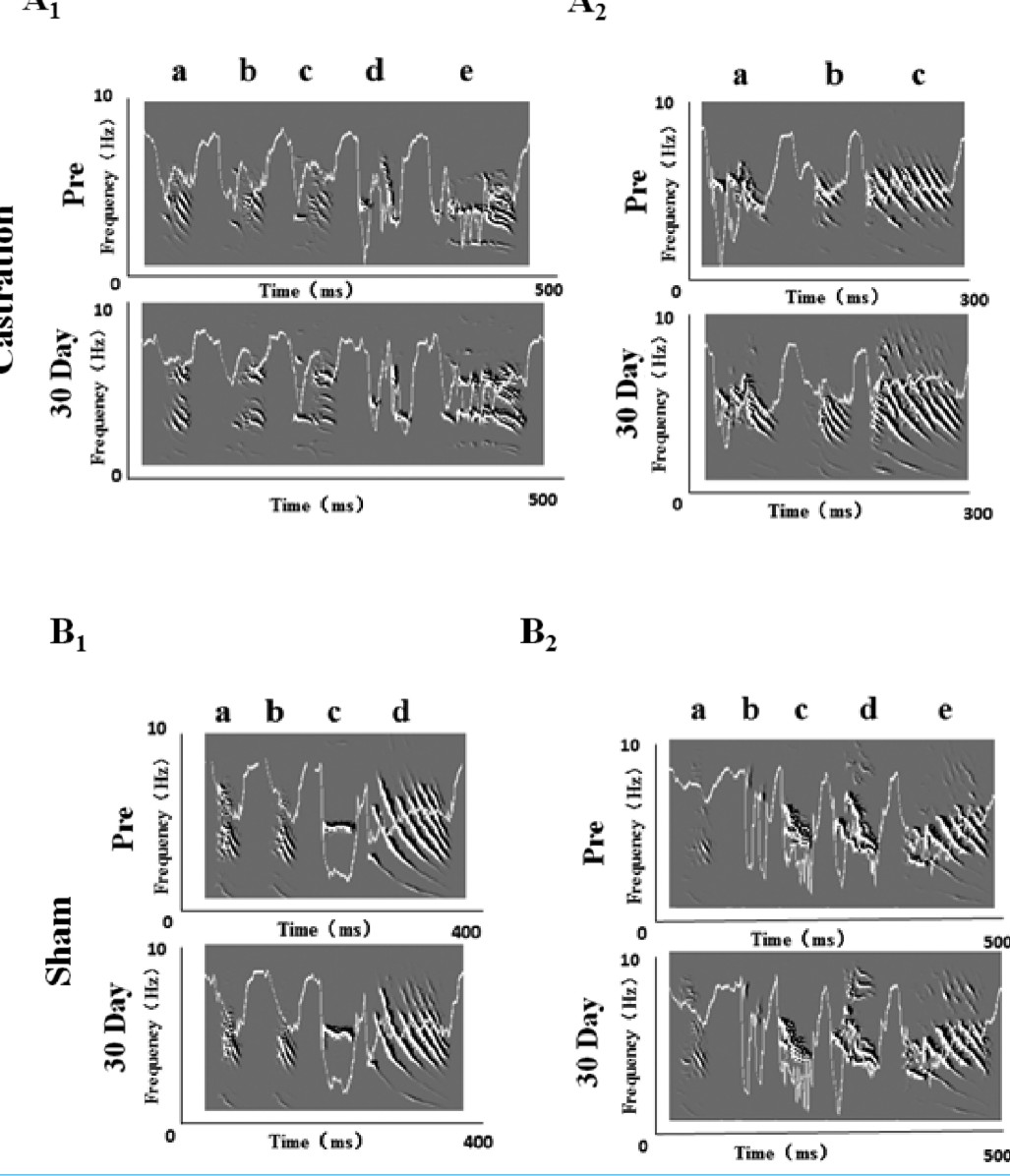

**Figure 1  Song sonograms and entropy curves (white line) of castration and sham groups in adult male zebra finches.** ($A_1, A_2$) The motifs of two birds in the castration group at "pre" operation and the 30th day after castration, respectively. ($B_1, B_2$) The motifs of two birds in the sham group at "pre" operation and the 30th day after sham operation, respectively. When white line became lower, the entropy of syllable was smaller.

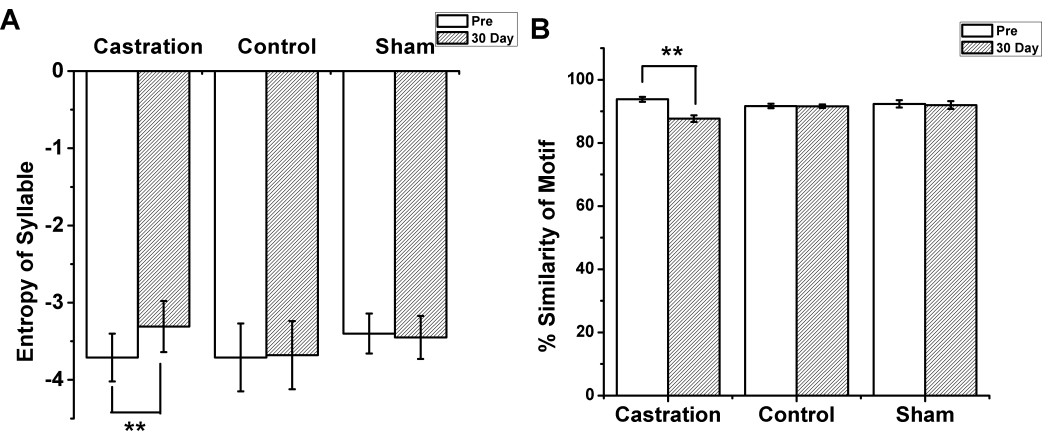

**Figure 2** The entropy of syllable and % similarity of motif in castration, control, and sham groups. (A) Castration group increased the entropy of syllables. Control and sham groups did not change. (B) Castration group exhibited decreased % similarity of motif. Control and sham groups showed no change.

**Table 1** The electrophysiological properties of RA PNs.

| Property | Castration | Control | Sham |
|---|---|---|---|
| Spontaneous firing rate (Hz) (cell-attached) | $8.13 \pm 1.00$ ($n = 21$) | $12.49 \pm 1.17^{**}$ ($n = 23$) | $11.32 \pm 0.74^{**}$ ($n = 10$) |
| Resting membrane potential (mV) | $-65.13 \pm 1.22$ ($n = 6$) | $-65.37 \pm 1.20$ ($n = 7$) | $-65.32 \pm 1.27$ ($n = 2$) |
| Spontaneous firing rate (Hz) (Whole-cell) | $6.58 \pm 0.89$ ($n = 15$) | $10.53 \pm 1.23^{*}$ ($n = 16$) | $10.97 \pm 1.26^{**}$ ($n = 8$) |
| Membrane time constant (ms) | $25.44 \pm 3.58$ ($n = 16$) | $43.98 \pm 4.42^{**}$ ($n = 16$) | $46.28 \pm 4.46^{**}$ ($n = 8$) |
| Input resistance (MΩ) | $197.75 \pm 10.25$ ($n = 16$) | $228.11 \pm 14.58$ ($n = 17$) | $226.98 \pm 17.33$ ($n = 8$) |
| Capacitance (pF) | $126.65 \pm 14.71$ ($n = 16$) | $194.08 \pm 20.97^{*}$ ($n = 16$) | $201.49 \pm 21.88^{**}$ ($n = 8$) |
| FI slope (Hz/pA) | $0.186 \pm 0.014$ ($n = 15$) | $0.296 \pm 0.017^{**}$ ($n = 13$) | $0.289 \pm 0.016^{**}$ ($n = 6$) |
| AP threshold (mV) | $-50.12 \pm 1.73$ ($n = 15$) | $-49.59 \pm 1.74$ ($n = 16$) | $-50.14 \pm 1.63$ ($n = 8$) |
| AHP peak amplitude (mV) | $-17.30 \pm 1.16$ ($n = 15$) | $-17.57 \pm 1.64$ ($n = 16$) | $-16.05 \pm 1.70$ ($n = 8$) |
| AHP time to peak (ms) | $21.38 \pm 1.53$ ($n = 15$) | $15.44 \pm 1.85^{**}$ ($n = 16$) | $15.03 \pm 2.03^{**}$ ($n = 8$) |
| Half-width (ms) | $1.10 \pm 0.10$ ($n = 15$) | $1.26 \pm 0.34$ ($n = 16$) | $0.99 \pm 0.13$ ($n = 8$) |
| Peak amplitude (mV) | $42.32 \pm 2.44$ ($n = 15$) | $49.17 \pm 2.69$ ($n = 16$) | $48.99 \pm 2.50$ ($n = 8$) |

**Notes.**

Numbers in parentheses indicate sample size. Resting membrane potential is only calculated for quiescent, non-spontaneously active neurons.

[*] $P < 0.05$.

[**] $P < 0.01$.

## Castration decreased spontaneous firing rates in the whole-cell configuration

In seasonally breeding songbirds, breeding conditions increase spontaneous firing rates in the whole-cell configuration (*Meitzen et al., 2009b*). In our experiment, castration significantly decreased spontaneous firing rate of RA PNs compared with the control group ($F_{(1,31)} = 6.65, P = 0.015$) and the sham group ($F_{(1,22)} = 8.26, P < 0.01$) (Table 1, Figs. 3D–3F).
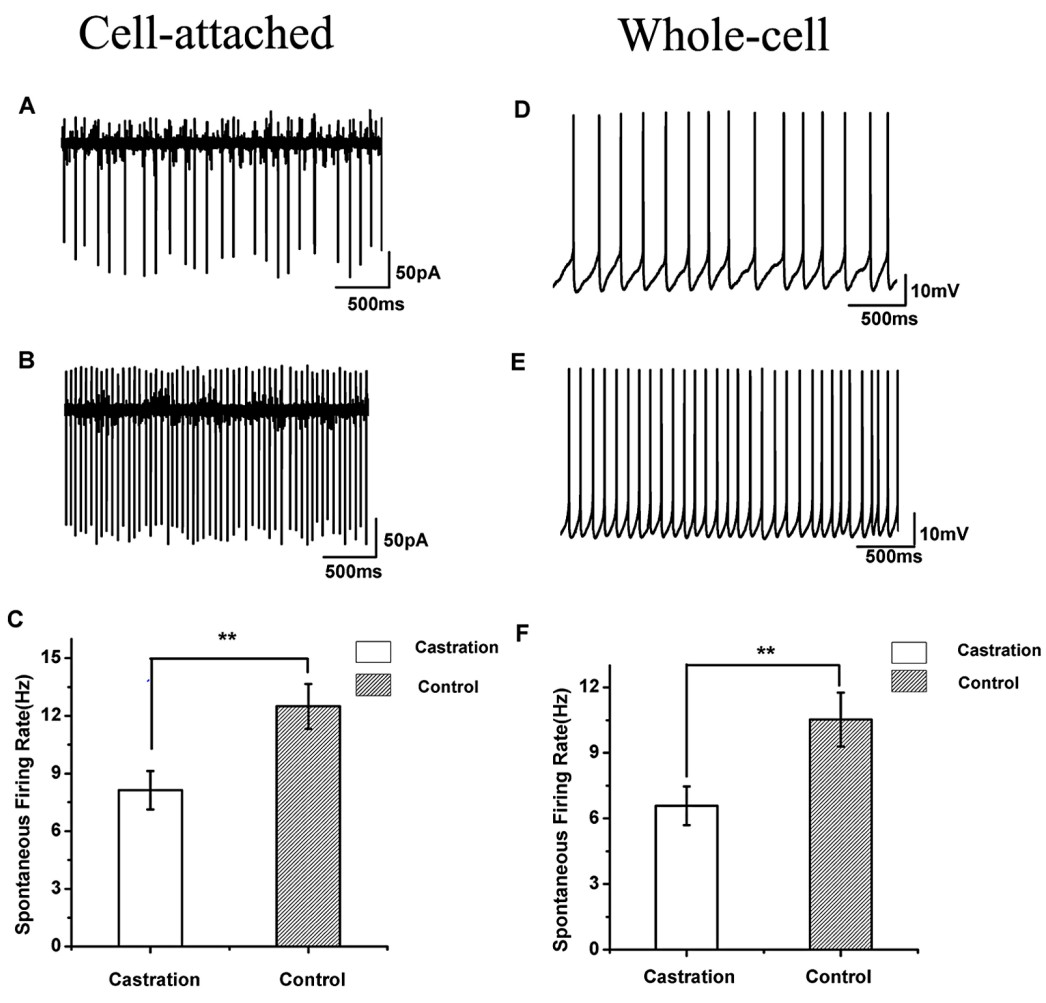

**Figure 3 The spontaneous firing of RA PNs in the cell-attached and whole-cell configuration.** (A, B) Example traces of spontaneous firing in RA PNs of the castration and control groups, respectively. (C) Castration significantly decreased spontaneous firing rates. (D, E) Example traces of spontaneous firing in RA PNs of the castration and control groups, respectively. (F) Castration significantly decreased spontaneous firing rates.

## Castration decreased evoked firing rates

AP firing rates evoked by depolarizing current injection were significantly decreased in the castrated group (Figs. 4A and 4B). At the current of 100 pA for 500 ms, the mean number of evoked firing was $13.63 \pm 1.12$ ($n = 15$) in the castrated group, $18.00 \pm 0.72$ ($n = 16$) in the control group and $17.88 \pm 0.88$ ($n = 8$) in the sham group. Castration significantly decreased the number of evoked firing compared with the control group ($F_{(1,31)} = 10.43, P < 0.01$) (Fig. 4C) and the sham group ($F_{(1,22)} = 5.76, P = 0.048$). When the currents were set from 0 to 200 pA for 500 ms at 50 pA steps and 10 s intervals, castration decreased the mean number of evoked firing, particularly at 50 pA, 100 pA, 150 pA, and 200 pA ($P < 0.01$). F–I curves were linearized. Castration significantly decreased the

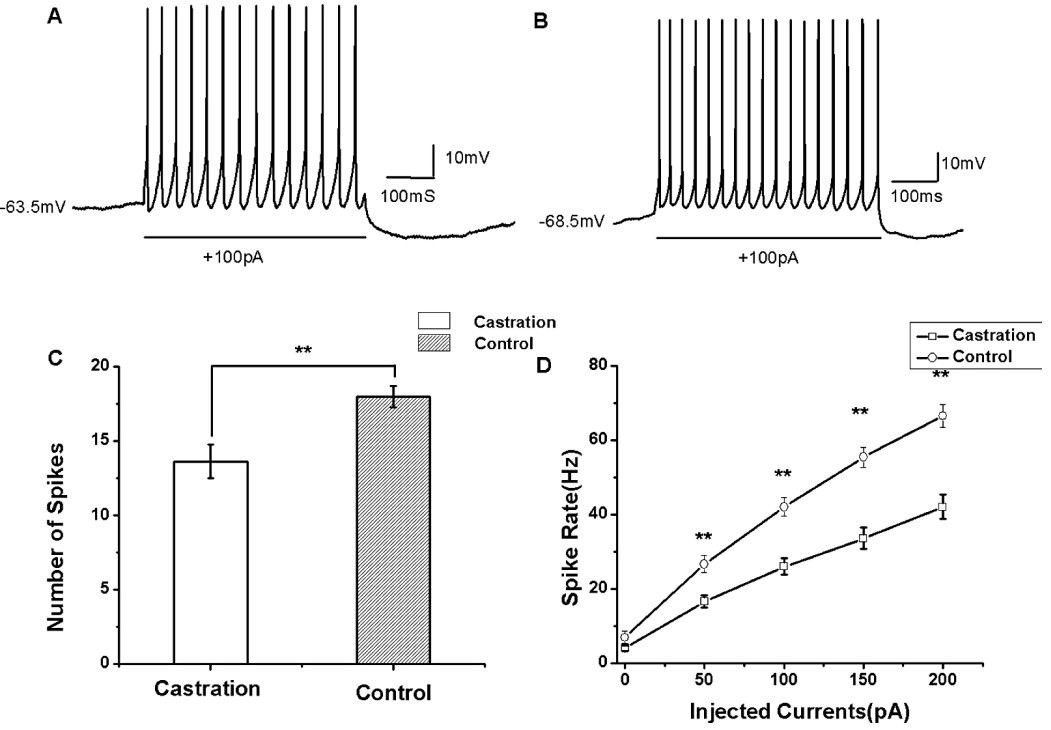

**Figure 4 The evoked firing of RA PNs in the whole-cell configuration.** (A, B) Example traces of AP firing in RA PNs of the castration and control groups in response to injecting a current of 100 pA for 500 ms, respectively. (C) Castration significantly decreased the number of evoked firing when injecting a current of 100 pA for 500 ms. (D) F–I curve of the castration and control groups. The slope of the F–I curve in the castration group was lower than that of the control group.

slope of F–I curve compared with the control group ($F_{(1,31)} = 21.87, P < 0.01$) (Fig. 4D) and sham group ($F_{(1,22)} = 27.89, P < 0.01$) (Table 1).

## Castration decreased the membrane time constant and capacitance, but increased AHP time to peak

To compare input resistance, currents from −200 to 20 pA for 500 ms at 10 pA steps and 10 s intervals were injected. The slope of the I–V curve by linear fit was the input resistance. There was no significant difference between the castrated and control groups. Castration significantly decreased the membrane time constant compared with the control group ($F_{(1,29)} = 10.62, P < 0.01$) (Fig. 5A) and the sham group ($F_{(1,21)} = 12.17, P < 0.01$). Castration also decreased membrane capacitance compared with the control group ($F_{(1,29)} = 6.93, P = 0.013$) (Fig. 5B) and the sham group ($F_{(1,21)} = 8.35, P < 0.01$). The intrinsic properties of spontaneous AP were also analyzed. AP threshold, AHP peak amplitude, half-width, and peak amplitude of the castration group were similar to the control and sham groups. However, castration significantly prolonged the AHP time to peak compared with the control group ($F_{(1,29)} = 12.76, P < 0.01$) (Figs. 5C and 5D) and the sham group ($F_{(1,21)} = 13.32, P < 0.01$) (Table 1).

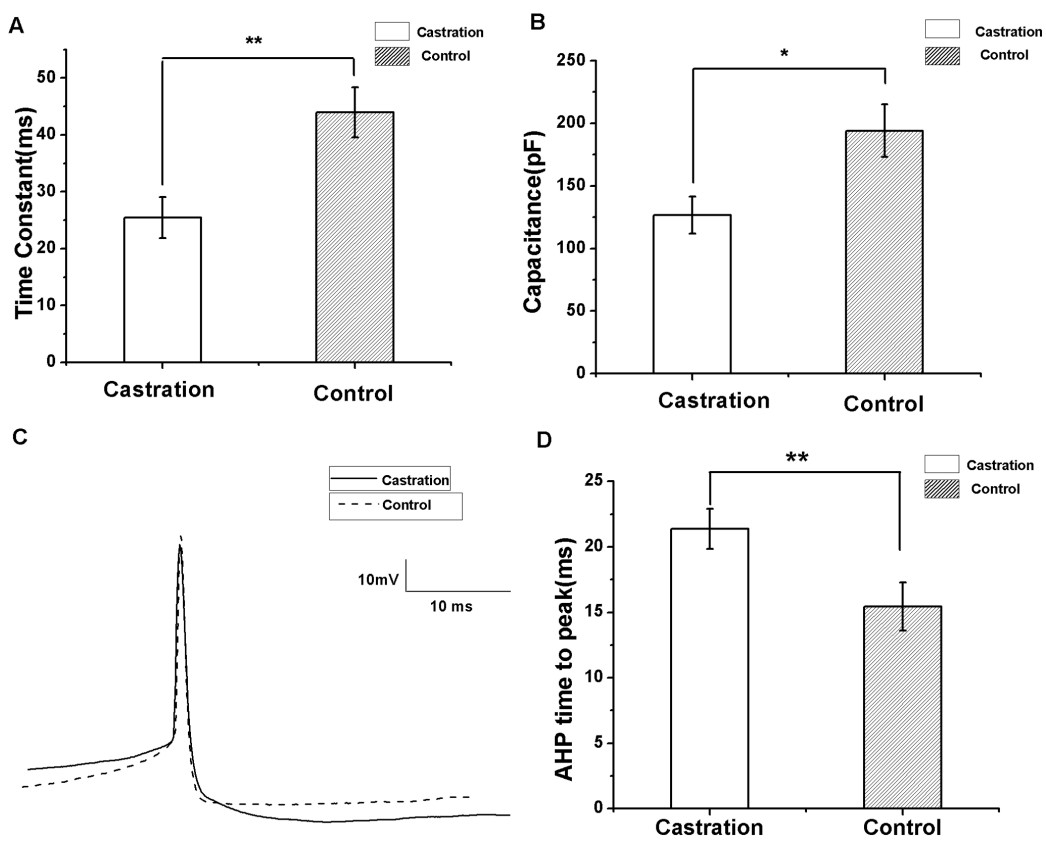

**Figure 5  The differences in membrane time constant, capacitance and AHP time to peak between the castration and control groups.** (A) Castration decreased the membrane time constant. (B) Castration decreased the membrane capacitance. (C) Example traces of AP in the castration and control groups, showing differences in the AHP time to peak. (D) Castration prolonged the AHP time to peak.

## DISCUSSION

In the present study, castration declined plasma testosterone levels in adult male zebra finches, although the decline was not as marked as those previously reported (*Adkins-Regan et al., 1990*; *Dloniak & Deviche, 2001*; *Luine et al., 1980*; *Marler et al., 1988*; *Strand & Deviche, 2007*). This difference may be due to residual testicular tissue in our study, the brain also makes steroids (*Schlinger & London, 2006*; *Schlinger & Remage-Healey, 2012*), or differences between breeding and non-breeding zebra finches.

High testosterone levels are associated with song stability, which reduces the potential for vocal plasticity (*Williams, Connor & Hill, 2003*). We found that castration increased the entropy of syllable and decreased the % similarity of motif in the songs, suggesting that castration decreased the stability of syllables and songs. Our result supported the effects that were described in William's (*Williams, Connor & Hill, 2003*). However, our results differ from the report of other previous described, in which castration exerts no effects on the stereotyped songs (*Arnold, 1975*; *Bottjer & Hewer, 1992*). It may be the difference of the software for analyzing birdsong. Like seasonally breeding songbirds, when the testosterone and its metabolite levels are high, the song nuclei and the syringes are fully

grown (*Brenowitz et al., 1998*; *Brenowitz et al., 1991*; *Hall & Macdougall-Shackleton, 2012*; *Tramontin, Hartman & Brenowitz, 2000*). Testosterone withdrawal induces the regression in the volume of HVC, RA and Area X (*Thompson, Bentley & Brenowitz, 2007*). The inactivation androgen and estrogen receptors in HVC prevent seasonal-like increases in song stereotypy (*Meitzen et al., 2007a*; *Meitzen & Thompson, 2008*). When testosterone levels decline, the expression of androgen receptors decrease in HVC (*Fraley et al., 2010*). Therefore, lower testosterone levels resulted in less stereotyped songs.

It was previously reported that testosterone and its metabolite estradiol can regulate the electrophysiological properties of RA PNs (*Meitzen et al., 2007a*; *Meitzen, Perkel & Brenowitz, 2007b*; *Meitzen et al., 2009b*). Castration induced lower plasma levels of testosterone and its estrogenic and androgenic metabolites. Lower levels of androgens and estrogens act transsynaptically within the HVC affecting the electrophysiological properties of RA PNs (*Meitzen et al., 2007a*). RA is an important premotor nucleus, and these changes in its intrinsic properties may directly modify the motor control of song production, resulting in changes in song stereotypy. In this study, castration decreased spontaneous and evoked firing rates, as well as the membrane time constant and capacitance, but increased AHP time to peak. These results indicate that castration decreased the excitability of the RA PNs, as previously reported (*Meitzen et al., 2009b*). Castration decreased the spontaneous firing rates and evoked firing rates, which reduce the ability of RA PNs to produce AP in response to synaptic input, particularly from HVC (*Meitzen et al., 2009b*). Castration decreased the membrane time constant, which might shorten the time available to integrate synaptic input (*Meitzen et al., 2009b*). As such, RA PNs slowly integrate relatively sparse inputs from the HVC to produce patterned firing that is closely correlated with song production (*Yu & Margoliash, 1996*). The decrease in membrane capacitance induced by castration may be related to the decrease of the size of RA PNs (*Meitzen et al., 2009b*). As previously described, testosterone withdrawal induced the regression in somatic area of RA neuron (*Thompson, Bentley & Brenowitz, 2007*). Castration also increased the AHP time to peak, which may be associated with the suppression of large conductance calcium-activated potassium channels (*Bean, 2007*; *Faber & Sah, 2002*).

The HVC-RA pathway contributes to stable song, while the LMAN-RA pathway contributes to variable song (*Kao, Wright & Doupe, 2008*; *Nottebohm, 2005*; *Olveczky, Andalman & Fee, 2005*). Testosterone-stimulated growth of the HVC is sufficient to induce growth of its efferent nucleus-RA (*Brenowitz & Lent, 2002*). High testosterone levels increase axonal density in the HVC-RA pathway (*De Groof et al., 2008*). In our castrated birds, low testosterone levels may decrease axonal density in the HVC-RA pathway. High testosterone levels decrease levels of NR2B mRNA, which is modulatory subunits of N-methyl-D-aspartic acid receptor (NMDAR), within the LMAN and the RA (*Singh et al., 2003*). The LMAN-RA input is largely mediated by the NMDAR (*Mooney & Konishi, 1991*). In our castrated birds, low testosterone levels may increase the modulation of the LMAN-RA input by the NMDAR. These effects may produce unstable songs.

Finally, the castration-induced decline in testosterone levels and excitability of the RA PNs may decrease the size of the song nuclei, and the synapses from the HVC to the RA, and increase the input from the LMAN to the RA. All of these effects would produce less stereotyped songs.

In conclusion, our study revealed that castration decreased song stereotypy and the excitability of RA PNs. The results provide a further understanding of how steroid sex hormones modulate electrophysiological properties to change song behavior.

### Funding

This work was supported by the National Natural Science Foundation of China (31172092). The funders had no role in study design, data collection and analysis, decision to publish, or preparation of the manuscript.

### Grant Disclosures

The following grant information was disclosed by the authors:
National Natural Science Foundation of China: 31172092.

### Competing Interests

The authors declare there are no competing interests.

### Author Contributions

- Songhua Wang conceived and designed the experiments, performed the experiments, analyzed the data, wrote the paper, prepared figures and/or tables.
- Congshu Liao performed the experiments, prepared figures and/or tables.
- Fengling Li and Shaoyi Liu performed the experiments.
- Wei Meng analyzed the data.
- Dongfeng Li conceived and designed the experiments, contributed reagents/materials/analysis tools, wrote the paper, reviewed drafts of the paper.

### Animal Ethics

The following information was supplied relating to ethical approvals (i.e., approving body and any reference numbers):

All experiments were carried out in accordance with the university and national animal guidelines. The care and use of animals reported on in this study were approved by the Institutional Animal Care and Use Committee at South China Normal University and in accordance with National Institutes of Health guidelines (scnu20070033).

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
