# Peer review of "Castration modulates singing patterns and electrophysiological properties of RA projection neurons in adult male zebra finches"

_PeerJ, doi:10.7717/peerj.352_

## Round 0.1 · original submission · Minor Revisions

The reviewers clearly offered many very helpful suggestions. Please address them fully in your revised manuscript.

Reviewer 1 ·

Basic reporting

Main comments:
In this study, the effects of castration on singing patterns and electrophysiological properties of projection neurons (PNs) in the robust nucleus of the arcopallium (RA) of adult male zebra finches were reported. It was fund that castration decreased the plasma levels of testosterone, increased song syllable’s entropy and reduced the similarity of motif. It was also demonstrated that the spontaneous and evoked activities in RA PNs were also reduced by castration. The results indicate that the excitability and activities of RA PNs and song stereotypy are related with the testosterone level in the body. The finding is interesting and is of scientific significance.

Minor points:
1. The figures were shown in a wrong order.
2. The figures are not matched the legends shown in the figures.

Experimental design

reasonable

Validity of the findings

The finding is interesting and is of scientific significance.

Reviewer 2 ·

Basic reporting

No comments

Experimental design

See below.

Validity of the findings

See below.

Additional comments

Review of “#2014:02:1465:0:1:REVIEW “ PeerJ
“Castration Modulates Singing Patterns and Electrophysiological Properties of RA Projection Neurons in Adult Male Zebra Finches”

This paper from Songhua Wang and colleagues is the latest in this laboratory’s study of the songbird pre-motor song control nucleus RA. Here they test the hypothesis that RA neuron electrophysiological properties will vary with reproductive state in the zebra finch. The experiment is straightforward, given that zebra finch males are opportunistic breeders who are essentially always in breeding state in typical laboratory housing. Thus the birds were castrated (or sham operated) and their RA neuron electrophysiological properties compared to birds with intact testes. Wang and colleagues find that RA neuron properties change dramatically, similar to previous experiments performed in seasonally-breeding birds. The experiments are well executed; however there are aspects of the manuscript which must be changed before publication.

Primary Comments

1. The most pressing issue is with the discussion of the role of testosterone. In the brain, testosterone can be converted to estradiol, which acts on estrogen receptors. In both developing zebra finches and seasonally breeding birds it is clear that testosterone converted to estradiol plays a strong role in mediating RA nucleus properties (including electrophysiological properties). In seasonally-breeding birds, androgen and estrogen receptors are activated in the song nucleus HVC must be activated to drive change in nucleus RA. There are androgen receptors in RA but their function is still mysterious. Another idea that should be addressed is that the zebra finch brain can produce androgens and estrogens independent of the testes, so circulating steroid hormone levels are not necessarily identical to that of the brain parenchyma (Schlinger and London, 2006). The authors work indicates that this is not playing much of a role regarding RA, but it should at least be mentioned. Thus much of the background and discussion needs to be re-written to reflect this body of previous work. One review that covers this is “Cell death and the song control system: A model for how sex steroid hormones regulate naturally-occurring neurodegeneration” by Christopher K. Thompson, Development, Growth & Differentiation, 53: 213-224, 2011.

2. Hormone test: It is odd that the testosterone levels for the castrated zebra finch were high compared to previous papers (i.e., Adkins-Regan et al., 1990, Gen & Comp Endocrinology and other papers as well), especially since the electrophysiological and song data argue that castration was effective. Did the authors double check after animal sacrifice that the testes were totally removed? The authors mention residual testes tissue on line 206 – did they actually see this or is this conjecture? What was the intra-assay variation in the samples? Also, in the assay did the substrate standards result in the curve recommended by the manufacturer?

Secondary comments

1. Reference numbers are missing in the main body of the paper.

2. Some of the classic papers on Zebra finch castration and testosterone are missing (i.e., Arnold, 1975; Cynx’s work on testosterone and changes in zebra finch song, Bottjer and Hewer, 2004)

3. Line 242: The wording of this sentence needs to be corrected. Photoperiod induces changes in the testes, which then releases testosterone., which then, along with estradiol, changes the song control system. There are also other actions of photoperiod, but it is incorrect to say that “androgens, rather than photoperiod, modulate electrophysiological properties to change song behavior.”

4. The figures seem to be out of order? This is probably an uploading issue.

5. Table 1. Resting membrane potential. If neurons are spontaneously active then there is not a resting membrane potential. If some of the neurons are quiescent, then that should be noted along with how many neurons are spontaneously active. Otherwise the correct entry would be “N/A” for “not applicable.” Also, the columns are out of alignment beginning with AHP peak amplitude.
6. Figure 5 C, D. The axis is mislabeled, it should be either spike rate (Hz) or number of spikes.

---

## Round 0.2 · accepted · Accept

Please make the further minor revisions as requested by reviewer 2.

Reviewer 2 ·

Basic reporting

Please see below.

Experimental design

Please see below.

Validity of the findings

Please see below.

Additional comments

Re-review, 2014:02:1465:1:1, PeerJ

The authors have made substantial revisions to the manuscript to address my comments. Pending some minor modifications (detailed below) I believe that this manuscript will be a nice addition to the literature.

Minor Concerns

Figure 5C: Missing scale bars

Table 1: In the table legend it should be noted that the “Resting membrane potential” was only calculated for quiescent, non-spontaneously active neurons.

Methods or Discussion: It needs to be stated that the authors did not double check that the testes where totally removed after animal sacrifice.